# Involvement of the p38/MK2 Pathway in MCLR Hepatotoxicity Revealed through MAPK Pharmacological Inhibition and Phosphoproteomics in HepaRG Cells

**DOI:** 10.3390/ijms241311168

**Published:** 2023-07-06

**Authors:** Katherine D. Lynch, Dayne T. Iverson, Namrata K. Bachhav, Michael Ridge Call, Guihua Eileen Yue, Bhagwat Prasad, John D. Clarke

**Affiliations:** Department of Pharmaceutical Sciences, Washington State University, Spokane, WA 99202, USA; katherine_lynch@wsu.edu (K.D.L.); bhagwat.prasad@wsu.edu (B.P.)

**Keywords:** microcystin-LR, phosphoproteomics, HepaRG, hepatocytes, p38 MAPK

## Abstract

Microcystin-leucine arginine (MCLR) is one of the most common and toxic microcystin variants, a class of cyanotoxins produced by cyanobacteria. A major molecular mechanism for MCLR-elicited liver toxicity involves the dysregulation of protein phosphorylation through protein phosphatase (PP) inhibition and mitogen-activated protein kinase (MAPK) modulation. In this study, specific pharmacological MAPK inhibitors were used in HepaRG cells to examine the pathways associated with MCLR cytotoxicity. SB203580 (SB), a p38 inhibitor, rescued HepaRG cell viability, whereas treatment with SP600125 (JNK inhibitor), MK2206 (AKT inhibitor), or N-acetylcysteine (reactive oxygen species scavenger) did not. Phosphoproteomic analysis revealed that phosphosites—which were altered by the addition of SB compared to MCLR treatment alone—included proteins involved in RNA processing, cytoskeletal stability, DNA damage response, protein degradation, and cell death. A closer analysis of specific proteins in some of these pathways indicated that SB reversed the MCLR-mediated phosphorylation of the necroptosis-associated proteins, the mixed lineage kinase domain-like protein (MLKL), receptor-interacting serine/threonine kinase 1 (RIP1), DNA damage response proteins, ataxia telangiectasia and Rad3-related kinase (ATR), and checkpoint kinase 1 (CHK1). Overall, these data implicate p38/MK2, DNA damage, and necroptosis in MCLR-mediated hepatotoxicity, and suggest these pathways may be targets for prevention prior to, or treatment after, MCLR toxicity.

## 1. Introduction

Microcystin-leucine arginine (MCLR) is one of the most common and toxic variants of microcystins, which are a class of cyanotoxins produced by freshwater cyanobacteria [1,2,3,4]. MCLR exposures occur through drinking water, recreation in water, and/or the consumption of contaminated fish and vegetables [1,2,3,4]. Evidence from preclinical models, case studies, and epidemiological studies indicate that the liver is the primary site of acute and chronic toxicity. Accidental acute exposure to MCLR through dialysis fluid resulted in the death of 60 patients due to severe liver toxicity, and recreational exposure to cyanotoxins in an infant resulted in the child requiring a liver transplant [3]. Regarding chronic toxicity, a case–control study in southwest China reported that serum MCLR concentrations were an independent risk factor for the development of hepatocellular carcinoma [5]. The International Agency for Research on Cancer has classified MCLR as a class 2B possible carcinogen [6]. To mitigate these risks, the World Health Organization has set a tolerable daily intake value of 0.04 µg/kg, and other governing agencies have set guidelines for MCLR in drinking and recreational water [7]. 

A major molecular mechanism for MCLR-elicited liver toxicity involves protein phosphatase (PP) inhibition and consequent mitogen-activated protein kinase (MAPK) dysregulation [2,3,8]. Protein phosphatases and kinases act in concert to coordinate protein phosphorylation as an important molecular mechanism to allow for rapid response to external stimuli. MCLR-mediated protein phosphatase inhibition occurs through the covalent binding of MCLR to the catalytic subunit of PP2A and PP1 [2,3,9]. MCLR toxicity is reported to activate MAPKs integral to cellular stress responses and cell survival/proliferation pathways, such as extracellular signal-regulated kinase (ERK) 1/2, ERK5, p38, c-Jun N-terminal kinase (JNK), and phosphatidylinositol 3-kinase (PI3-K/AKT) [3,9,10,11,12]. For example, MCLR decreased PP2A activity and increased phosphorylation of AKT, ERK1/2, JNK, and p38 in mouse livers as early as 2 h post-exposure [12]. Although HepG2 liver cells were insusceptible to decreases in MCLR-induced viability of up to 1 µM, MCLR doses ranging from 0.001 to 1 µM induced ERK phosphorylation but had no effect on p38 or JNK phosphorylation [10]. Another human liver cell line, HL7702, was susceptible to viability changes at or above MCLR doses of 5 µM, and ERK phosphorylation occurred at 10 and 20 µM MCLR exposure [11]. Komatsu et al. reported a time-dependent induction of phosphorylated ERK1/2, JNK, and p38 in HEK293 cells overexpressing organic anion-transporting polypeptide (OATP) 1B1 and OATP1B3 [8]. Although these data clearly connect MCLR exposure with the induction of MAPK pathways, to date, a thorough investigation concerning the critical importance of MAPK pathways for MCLR-mediated cytotoxicity has not been conducted. In the current study, specific pharmacological MAPK inhibitors were used in HepaRG cells, a versatile human liver cell line that is functionally similar to primary hepatocytes [13,14,15], to examine the MAPK pathways associated with MCLR toxicity. In addition, phosphoproteomic analysis was performed after MCLR exposure in the presence and absence of a MAPK inhibitor to elucidate specific pathways important for MCLR-mediated hepatocyte death. 

## 2. Materials and Methods

### 2.1. Chemicals and Reagents

William’s E. basal medium (WEM) (cat. 112-033-101) was purchased from Quality Biological (Gaithersburg, MD, USA). L-glutamine (cat. TMS-002-C) and N-acetyl-L-cysteine (NAC) (cat. A9165) were purchased from Millipore Sigma (Burlington, MA, USA). Penicillin and streptomycin solution (cat. 25-512) was purchased from Genesee Scientific (San Diego, CA, USA). Dimethyl-sulfoxide (DMSO) (cat. 36480-AP), protease and phosphatase inhibitor cocktail (cat. 78440), dithiothreitol (cat. D1532), iodoacetamide (cat. A39271), ammonium bicarbonate (cat. 393210050), and trypsin (cat. 90057) were purchased from Thermo Fisher Scientific (Waltham, MA, USA). MCLR (cat. 10007188, ≥95% purity) was purchased from Cayman Chemicals (Ann Arbor, MI, USA). 3-(4,5-Dimethylthiazol-2-yl)-2,5-diphenyltetrazolium bromide (MTT) (cat. 142015) was purchased from BeanTown Chemical (Hudson, NH, USA). SP600125 (SP) (JNK inhibitor, cat. HY-12275), MK2206 di-hydrochloride (MK) (AKT inhibitor, cat. HY-10358), and adezmapimod hydrochloride, SB203580 (SB) (p38 inhibitor, cat. HY-10256A) were purchased from Med Chem Express (Monmouth Junction, NJ, USA).

### 2.2. Cell Culture

Undifferentiated HepaRG cells (passage 14) (cat. HPR101) were purchased from BioPredic International (Saint Grégoire, France) and cultured as per manufacturer’s protocols in a humidified atmosphere containing 5% CO_2_ at 37 °C. Briefly, undifferentiated HepaRG cells were cultured in WEM with growth supplement (cat. ADD710, BioPredic), 100 μg/mL penicillin/streptomycin, and 2 mM L-glutamine for 14 days before being differentiated for an additional 14 days in WEM with differential supplement (cat. ADD720, BioPredic). Media were renewed every 2 or 3 days. Cells were seeded at 9.0 × 10^3^ cells per well in 96-well plates, or 2.0 × 10^5^ cells per well in 12-well plates. All experimental treatments were performed on differentiated HepaRG passages 16–20 in the differential media.

### 2.3. Viability Assays

Cells were incubated in vehicle (0.5–1% DMSO) with or without pharmacological compounds (MK, SB, NAC, SP) for 1 h. Cells were then incubated with MK, NAC, and SP in the presence of MCLR for 24 h, whereas SB was incubated with MCLR for 24, 48, and 72 h. Non-cytotoxic concentrations of pharmacological compounds were selected based on known pharmacologically active doses from the literature [16,17,18] and our dose response screening in HepaRG cells. Following the incubation period, cells were washed twice with PBS and then exposed to differential media with 0.5 mg/mL MTT for 2 h in humidified atmosphere containing 5% CO_2_ at 37 °C. Cells were lysed in 100% DMSO and absorbance was measured at 580 nm using a Synergy H1 (BioTek, Winooski, VT, USA) plate reader. The lethal MCLR concentration that killed 50% of the cells (LC_50_) at each time point was calculated in GraphPad Prism 7 software (GraphPad software, INC., La Jolla, CA, USA) using an inhibitor concentration versus normalized response equation with a variable hillslope. 

### 2.4. Immunoblot Assays

Cells were washed with PBS twice prior to the addition of NP-40 lysis buffer and freezing at −80 °C. Proteins were extracted after 10 min at 10,000× *g* at 4 °C. Protein concentrations were determined using the Pierce BCA Protein Quantification Assay kit (Thermo Fisher Scientific, cat. 23225). Ten to twenty micrograms of total protein were loaded into 7.5% or 10% SDS-PAGE gels and transferred to polyvinylidene difluoride membranes with a Bio-Rad (Hercules, CA, USA) Trans-Blot Turbo system at 25 V/1.0 A for 30 min. Membranes were blocked with 5% non-fat dry milk in Tris-buffered saline/Tween 20 (TBST) for 1 h at room temperature and incubated at 4 °C overnight in TBST-5% BSA 1:1000 dilution with the following primary antibodies from Cell Signaling (Boston, MA), unless otherwise indicated: p38 MAPK (cat. 9212); phospho-Thr180/Tyr182-p38 MAPK (phos-p38) (cat. 4511); receptor-interacting serine/threonine kinase 1 (RIP1) (cat. 3493); phospho-Ser166-RIP1 (pRIP1) (cat. 65746); phospho-Ser428-ATR (pATR) (cat. 2853); phospho-Thr68-CHK2 (pCHK2) (cat. 2197); phospho-Ser345-CHK1 (pCHK1) (cat. 2348); UV-damaged DNA binding protein1 (DDB1) (cat. 6998); damage-specific DNA binding protein 2 (DDB2) (cat. 5416); ring-box 1, also known as the regulator of cullins 1 (RBX1) (cat. 11922); mixed lineage kinase domain-like protein (MLKL) (cat. 14993); phospho-Ser358-MLKL (pMLKL) (cat. 91689), MAPKAPK-2 (MK2) (cat. 12155); phospho-Thr334-MK2 (pMK2) (cat. 3007); S-phase kinase-associated protein 1 (SKP1) (cat. 12248); cullin 4A (CUL4A) (cat. 2699); CYLD lysine 63 deubiquitinase (CYLD) (cat. 12797); ubiquitin (cat. 3936); MCLR (1:2000; cat. A1x-804-320-C200, Enzo, Farmingdale, NY, USA); and PP2A (1:2000; cat. 05-421, Millipore Sigma). The blots were incubated with respective secondary HRP-linked anti-mouse (cat. 20-304, Genesee Scientific) or anti-rabbit (cat. 20-303, Genesee Scientific) antibodies in 5% non-fat dry milk in TBST for 1 h at room temperature at 1:10,000 dilution. Blots were developed with SuperSignal West Pico or Femto (Thermo Fisher Scientific) and images captured using a Bio-Rad ChemiDoc imager. Densitometry was performed using Image Lab (Bio-Rad, Standard Edition, Version 6.0.0 build 25). All proteins of interest were normalized to total protein stained with amido black. In the case of modified proteins analyzed with total proteins, all bands of interest are normalized to respective amidos before comparison with total protein of interest. Total protein normalization is a commonly accepted technique for protein densitometry analysis instead of the single-protein loading control [19]. 

### 2.5. Phosphoproteomics

Fully differentiated HepaRG 8 × 10^6^ cells were treated for 1 h with vehicle or SB, then treated with MCLR (or vehicle) with and without SB for 24 h. Cells were washed with PBS twice, centrifuged, and the pellet was collected for phosphoproteomic analysis.

#### 2.5.1. Protein Digestion and Phosphopeptide Enrichment 

The cell pellet was lysed in urea buffer (8 M urea and 0.1 M ammonium bicarbonate with 1X protease and phosphatase inhibitor cocktail) in ice for 30 min. The lysed cells were centrifuged at 16,000× *g* at 4 °C for 15 min, and the supernatant was collected. An amount of 500 µg protein was reduced with 25 mM dithiothreitol at 37 °C for 1 h. The sample was then alkylated with 60 mM iodoacetamide at room temperature in the dark for 30 min Ammonium bicarbonate (0.1 M) was added to dilute the urea buffer to 1.6 M before digestion at room temperature for overnight with trypsin using trypsin/protein at a ratio of 1:40. All the digested samples were desalted with HLB solid-phase extraction cartridge (Waters, Milford, MA, USA) and dried with speed vacuum concentrator for phosphopeptide enrichment. 

Immobilized metal affinity chromatography (IMAC) was used for phosphopeptide enrichment. Ni-NTA agarose (Invitrogen, Waltham, MA, USA) was washed with water and treated with 0.1M EDTA for 30 min. The depleted metal ion beads were further washed with water and chelated with 0.1 M ferric chloride for 1 h. Dried desalted peptides were resuspended in 80% acetonitrile with 0.1% trifluoroacetic acid and mixed with charged immobilized metal affinity chromatography beads (50 µL resuspend beads with 100 µg sample ratio). Sample mixture was incubated for 1 h at room temperature and washed. The phosphopeptides were eluted using the elution buffer and dried with a speed vac for liquid chromatograph–mass spectrometry (LC-MS) analysis.

#### 2.5.2. LC-MS Analysis 

LC separation was performed using Easy-nLC 1200 (Thermo Fisher Scientific) coupled with Q Exactive HF Hybrid Quadrupole-Orbitrap mass spectrometer (Thermo Fisher Scientific). Peptides were separated using Pepmap RSLC C18 25 cm × 75 µm column (2 µm, 100 Å) with a 131 min gradient: 0–10 min: 2–6% buffer B (A: 0.1% formic acid, B: 80% acetonitrile with 0.1% formic acid), 10–110 min: 6–30% B, 110–115 min: 30–100% B, and 115–131 min: 100% B. The flow rate was set at 300 nL/min. MS analysis was performed with a spray voltage of 1.7 kV. The phosphopeptide analysis was performed with data-dependent acquisition mode with the following setting: top 10 ions, MS1 resolution 120,000, and MS2 resolution 30,000. LC-MS data were searched using Maxquant software with the following settings: fixed modification included carbamidomethylation, whereas the variable modification included oxidation (M), acetyl (protein N-term), and phosphorylation (STY). Proteins were identified with 1% false discovery rate. 

#### 2.5.3. Phosphosite Analysis

Phosphosite analysis was performed with RStudio version 2022.2.2.485 using the Bioconductor package PhosR [20,21]. Code for data processing and analyses was adapted from the PhosR package vignette and published code [20]. MaxQuant data were collected as described in Section 2.5.2 and used as input data for PhosR. Phosphosites were filtered to retain those with at least 50% quantification rate in one or more treatments. Imputation for missing values was performed using scImpute, which imputes missing values for a specific phosphosite across replicates within a treatment group. Data were log2-transformed and centered on the median. Differential phosphosite levels were determined using a linear model fit with adjusted *p*-value greater than 0.05 and fold change greater than 1.6 or less than −1.6. Kinase-substrate relationships from PhosphoSitePlus were used to predict the major kinases perturbed by certain treatments. Gene set enrichment analysis was performed using the Reactome database to identify pathways that were dysregulated due to either MCLR or SB treatment. 

### 2.6. Statistical Analysis

All data, except for the differential expression data, are represented as mean ± SD and were analyzed by two-way analysis of variance (ANOVA) or Student’s *t*-tests. Two-way ANOVA *p*-values are shown in tables accompanying the respective graphs. Sidak’s multiple comparison post-test were completed to determine the statistical differences due to p38 inhibition for vehicle and MCLR treatments. Data represent three independent experiments. All analyses were performed using GraphPad Prism 7 software.

## 3. Results

### 3.1. MCLR-Elicited Viability Changes and Pharmacological Intervention in HepaRG Cells

MCLR decreased cell viability in a dose- and time-dependent manner (Figure 1A). The lowest toxic dose at 24 h was 1 μM, which decreased to 0.01 μM at 48 h and 72 h. MCLR doses at or above 30 μM decreased viability by more than 50%. MCLR LC_50_ values were 13.6 ± 3.0, 4.4 ± 1.2, and 2.3 ± 0.5 µM at 24, 48, and 72 h, respectively (LC_50_ ± S.E.), although these data should be interpreted with caution because HepaRG cells comprise different cell types when fully differentiated [13]. N-acetyl-cystine (NAC), a JNK inhibitor, and an AKT inhibitor did not prevent MCLR-induced death (Figure 1B–D). SB at 100 μM completely rescued viability up to 10 μM MCLR at 24 and 48 h; however, SB did not completely rescue viability at 72 h (Figure 1E–G).

### 3.2. Phosphoproteomics Analysis after MCLR Toxicity and/or p38 Inhibition

MCLR dysregulated 1120 mono-phosphosites, 94.4% of which were upregulated, whereas SB dysregulated 1170 mono-phosphosites, 99.8% of which were downregulated (Figure 2A,B). The combination treatment of MCLR and SB dysregulated 1015 mono-phosphosites, a majority of which were upregulated (Figure 2C). Among the upregulated mono-phosphosites.

Across all treatments, 80.4% were shared between the two MCLR treatments (i.e., MCLR alone and MCLR with SB) (Figure 2D). Among the downregulated mono-phosphosites across all treatments, 4.7% were shared between the two SB treatments (i.e., SB alone and MCLR with SB) (Figure 2E). A Reactome analysis of the individual effects of MCLR alone and SB alone on mono-phosphosites revealed that only ~11% of the pathways were shared between the treatments (Table 1). MCLR altered pathways related to apoptosis, ERK/MAPK, JAK-STAT, and PPAR-α, and SB altered pathways related to chromosome maintenance, DNA damage, and MAPKs. All the dysregulated mono-phosphosites and di-phosphosites sites are listed in Appendix A. Among the upregulated di-phosphosites across all treatments, 14% were shared between the two MCLR treatments (i.e., MCLR alone and MCLR with SB). Among the downregulated di-phosphosites across all treatments, 8% were shared between the two SB treatments (i.e., SB alone and MCLR with SB) and 7% were shared between the MCLR alone and SB alone treatments. No Reactome pathways were identified for di-phosphosites.

When comparing the potential rescue effect of SB (i.e., MCLR with SB versus MCLR alone), there were 80 dysregulated mono-phosphosites (Figure 3A and Table 2). The 16 upregulated sites all overlapped with the MCLR treatments, while the 64 downregulated sites all overlapped with the SB treatments (Figure 3B,C). Kinase perturbation analysis identified MK2 as a kinase that was regulated in opposite directions by MCLR alone versus MCLR with SB (Figure 3D).

### 3.3. Mechanisms of p38 Inhibition for Rescue of MCLR Toxicity

#### 3.3.1. Protein Phosphatase Expression, MCLR Protein Binding, and p38/MK2 Phosphorylation 

MCLR decreased PP2A protein expression and increased protein-bound MCLR at 25 kDA, but SB did not significantly rescue these MCLR effects (Figure 4A,B). In contrast, two higher-molecular-weight MCLR-bound proteins at 45 kDa and 55 kDa were partially rescued by SB (Figure 4C,D). The specific proteins that were MCLR-bound, represented by these different molecular weights, were not determined here, although published evidence indicates MCLR binds PP1, PP2A, and ATP synthase [9,22,23]. MCLR increased the phosphorylation of p38 at Thr180/Try182 and MK2 at Thr334, and SB reduced phosphorylation of these sites back to baseline levels (Figure 4E,F). 

#### 3.3.2. Necroptosis, DNA Damage, and Protein Degradation Responses

MCLR increased phosphorylation of MLKL at Ser358, and SB reduced phosphorylation of this site to near baseline levels (Figure 5A). Phosphorylation of RIP1 at Ser166 was rescued by SB back to baseline (Figure 5B). Total CYLD expression was not affected by MCLR treatment (Figure 5C). MCLR increased phosphorylation of ATR at Ser428 and CHK1 at Ser345, and SB reduced phosphorylation of these sites near baseline levels (Figure 5D,E). MCLR treatment modestly increased phosphorylation of CHK2 at Thr68 but was not affected by SB (Figure 5F). 

MCLR did not affect RBX1 and SKP1 protein expression (Figure 6A,B), whereas MCLR increased the total levels of ubiquitinated proteins and decreased CUL4A expression (Figure 6C,D). SB treatment did not affect these MCLR effects on RBX1, SKP1, protein ubiquitination, and CUL4A expression. The total expression of DDB2 was not affected by MCLR (MCLR *p*-value 0.4953; SB203580 *p*-value 0.5466; interaction *p*-value 0.6204). A higher-molecular-weight-modified form of DDB2 increased after MCLR treatment, and SB reduced this MCLR effect (Figure 6E). MCLR treatment did not affect DDB1 protein expression. 

## 4. Discussion

The MCLR-mediated inhibition of protein phosphatases is known to alter MAPK pathways and affect cell survival. A limited number of publications have investigated which MAPK pathways are key in MCLR cytotoxicity in mammalian species. One previous publication reported a complete rescue of MCLR toxicity in HEK293 cells at 72 h with SB (p38 inhibitor), whereas SP600125 (JNK inhibitor) partially rescued viability [8]. In the current study, co-treatment with the p38 inhibitor (SB) completely rescued MCLR-mediated cell death up to 10 μM MCLR through 48 h and partially rescued MCLR toxicity at 72 h, whereas the 24 h MCLR co-treatment with SP600125 and MK2206 (AKT inhibitor) did not rescue HepaRG cell viability. These data suggest that p38 may be important in MCLR-mediated cell stress and led us to further investigate p38 pathways that may be responsible for the rescue of viability in HepaRG cells. 

Phosphoproteomic analysis was performed to determine the effects of MCLR and the combination of MCLR with the p38 inhibitor on the phosphoproteome. MCLR-mediated protein phosphatase inhibition in HepaRG cells increased the phosphorylation status of many proteins, and dysregulated Reactome pathways associated with metabolism (e.g., glycogen breakdown, lipid metabolism), inflammation (e.g., interleukin-12 signaling), and cell death (e.g., apoptotic execution phase). Although the p38 inhibitor alone had the opposite effect on the phosphoproteome (i.e., it decreased phosphorylation at many phosphosites), it was not able to overcome most of the increased protein phosphorylation caused by MCLR. The 80 phosphosites that were altered by the addition of the p38 inhibitor compared to MCLR treatment alone included proteins involved in RNA processing, cytoskeletal stability, DNA damage response, protein degradation, and cell death (apoptosis/necroptosis). These pathways are also related to the major p38-mediated cellular responses, including cell cycle regulation, DNA damage repair, and mRNA processing [24,25,26,27,28]. To our knowledge, this is the first phosphoproteomic report with MCLR and a p38 inhibitor, although a previous study treated HL7702 with MCLR and SB and observed a rescue of MCLR-mediated changes in both heat shock protein 27 and Tau phosphorylation [29,30]. These two proteins were not captured in our dataset possibly due to detection limits with untargeted phosphoproteomics. Further investigation into the potential rescue mechanism for the p38 inhibitor indicated that MK2 was the only kinase rescued by the p38 inhibitor during MCLR toxicity. MK2 is downstream of p38 and is important in multiple cellular processes, including cell stress response and regulation of cell viability through necroptosis. 

Necroptosis is a specialized form of cell death that is characterized by insertion of MLKL into the plasma membrane. MK2 phosphorylates RIP1, which phosphorylates MLKL and promotes its oligomerization and membrane insertion [31,32,33]. In HepaRG cells, MCLR increased phosphorylation of MK2, RIP1, and MLKL, and this increase was blocked by p38 inhibition. A previous study reported that MCLR increased protein levels of RIP1, RIP3, and MLKL and phosphorylated MLKL at 48 h in primary mouse hepatocytes. Treatment of the mouse hepatocytes with Nec-1, a necroptosis inhibitor, blocked the induction of these proteins [34]. It is not clear what impact changes in total RIP1 and MLKL protein may have on necroptosis, but in the current study, the expression of these proteins was unchanged. The MCLR-mediated induction of, and subsequent p38-mediated rescue of, MK2, RIP1, and MLKL phosphorylation indicate that MCLR induced necroptosis through the p38/MK2 pathway. Another important protein in necroptosis is CYLD, which is a deubiquitinase that removes ubiquitin from RIP1 and allows it to initiate the formation of the necrosome [31]. MCLR exposure did not change total CYLD protein levels in HepaRG cells, supporting the conclusion that MCLR primarily affected protein phosphorylation rather than protein turnover related to necroptosis. Collectively, these data indicate that necroptosis may be an important aspect of MCLR-mediated hepatotoxicity, although more experiments with necroptosis inhibitors and/or gene knockdown are required to confirm this. In addition, our data do not preclude the possibility that apoptosis or other cell death pathways may be involved in MCLR hepatotoxicity, as suggested by the identification of apoptosis in the Reactome pathway analysis. 

Oxidative stress and DNA damage are interrelated mechanisms that are often implicated in MCLR hepatotoxicity, both of which can also activate the p38 pathway [2,24,25,26,27,35]. Although many publications cite oxidative stress as an important mechanism in MCLR toxicity, the evidence using the antioxidant NAC is inconsistent and unclear. Indeed, in the current study, NAC treatment did not rescue MCLR cytotoxicity in HepaRG cells. These results contrast with previous data where NAC partially rescued the viability of OATP1B1/OATP1B3 overexpressing HEK293 cells [8]. Another study performed co-treatment of CHO cells with MCLR and NAC, and showed that NAC attenuated the MCLR-mediated increase in reactive oxygen species levels but did not rescue cell viability [18]. In contrast, NAC treatment in ICR mice blocked the increase in ALT and AST after a single dose of 55 µg/kg MCLR at 6 and 12 h [36]. It is important to note that NAC is a hydrophilic antioxidant that will not block all ROS generation. Previous data in primary mouse hepatocytes indicated that β-carotene reduced MCLR toxicity, whereas retinoids were weakly protective, and α-tocopherol did not protect the cells [37]. Thus, although oxidative stress may be involved in MCLR-mediated cell death [35], data from the current study indicate that NAC could not block MCLR-mediated HepaRG death. Additional experiments are required to determine whether lipophilic antioxidants (e.g., β-carotene, retinoids, or α-tocopherol) may modulate MCLR-mediated HepaRG death.

Although various types of DNA damage have been reported after MCLR-mediated toxicity (e.g., DNA double-strand breaks, oxidized pyrimidines/purines, micronuclei) [38,39,40,41,42], evidence for the specific DNA repair pathway(s) involved has been limited. In the current study, MCLR exposure to HepaRG cells had minimal effect on CHK2 but clearly activated the ATR/CHK1 pathway. The ATR/CHK1 pathway is activated by a broad range of DNA damage, including single- and double-strand breaks and DNA lesions [43]. Previous publications reported MCLR-elicited DNA double-strand breaks in HepG2, human peripheral lymphocytes, and HL7702 cells, as evident by an increased phosphorylated H2A histone family member X (γH2AX) or the comet assay [38,41,42]. The ATR/CHK1 pathway can be activated by replication stress [44], and CHK1 can directly phosphorylate and activate aurora kinase B (AURKB) to prevent the formation of lagging chromosomes [45]. DNA double-strand breaks and lagging chromosomes can result in the formation of micronuclei [46] and previous publications reported an increased micronuclei formation after MCLR exposure for 24 h in HL7702 and TK6 cells [40,42]. Our data support a role of ATR/CHK1/AURKB response and repair pathways in MCLR hepatotoxicity as indicated by an increased CHK1 phosphorylation and predicted an upregulation of AURKB. It is noteworthy that a recent publication reported DNA damage and ATM/p53/CHK2 activation after repeated MCLR exposure which contrasts with our results indicating a strong induction of CHK1 rather than CHK2 [47]. These data support the potential role of DNA double-strand breaks, replication stress, and/or chromosomal attachment and segregation problems after MCLR hepatotoxicity.

DDB2 is another crucial DNA damage response protein that recognizes many types of DNA damage [48]. After binding to the site of DNA damage, DDB2 forms a complex with CUL4A-DDB1, which ubiquitinates DDB2, causing it to dissociate from the DNA and allow repair complexes to access the site of DNA damage [49]. Oxidative stress is reported to increase DDB2 transcription through activated p38 [50], but there are no observable changes in total DDB2 protein expression 24 h after MCLR treatment herein. p38 is also reported to phosphorylate DDB2, which modulates DDB2-mediated chromatin remodeling during DNA damage [51]. Although the exact p38 target phosphosites are not known, our phosphoproteomic data demonstrated that MCLR increased DDB2 phosphorylation at serine 24 and serine 26 compared to vehicle, whereas co-treatment with SB decreased phosphorylation at these two sites, compared to MCLR alone. In addition, MCLR treatment decreased CUL4A protein expression, increased the amount of ubiquitinated proteins, and increased the amount of a larger molecular-weight-modified form of DDB2. Reactome analysis also indicated that pathways related to chromosome maintenance and DNA damage bypass were modulated by SB treatment. Taken together, these data support a role of CUL4A-DDB1-DDB2 in DNA damage response after MCLR exposure, but long-term MCLR exposure and/or recovery studies are needed to clarify whether the rescue of MCLR toxicity through p38 inhibition permits the perpetuation of DNA damage through the downregulation of the DNA damage response.

## 5. Conclusions

The dose- and time-dependent rescue of MCLR toxicity in HepaRG cells by the p38 inhibitor appears to be driven by a MK2-mediated decrease in necroptosis. The underlying mechanism for necroptosis inhibition could lie in the modulation of the DNA damage response pathways affected by p38 inhibition. Although a robust rescue in cell viability was observed, the overall differences in the phosphoproteome between MCLR alone and MCLR with SB were less marked, suggesting that the distinct pathways identified are more important than global changes in the phosphoproteome for MCLR cytotoxicity. Future investigations should include determining the long-term implications of modulating DNA damage repair pathways after MCLR exposure. 

## Figures and Tables

**Figure 1 ijms-24-11168-f001:**
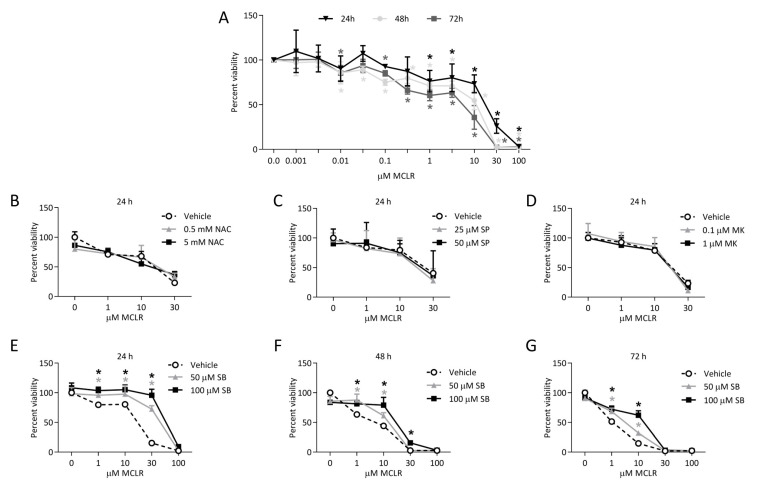
HepaRG viability after treatment with MCLR and/or pharmacological compounds. HepaRG cell viability after treatment with a range of MCLR concentrations for 24, 48, and 72 h (**A**). HepaRG cell viability after treatment with 0, 1, 10, or 30 μM MCLR and two concentrations of N-acetyl-cystine (NAC) (**B**), SP600125 (SP) (**C**), or MK2206 (MK) (**D**) for 24 h. HepaRG cell viability after treatment with 0, 1, 10, 30, or 100 μM MCLR and two concentrations of SB203580 (SB) for 24 h (**E**), 48 h (**F**), and 72 h (**G**). Error bars represent mean ± standard deviation of *n* = 3. * *p* < 0.05 compared to vehicle (**A**) and * *p* < 0.05 compared to respective MCLR treatment without pharmacological compounds (**B**–**G**). The color of each (*) corresponds to the respectively color for each treatment time.

**Figure 2 ijms-24-11168-f002:**
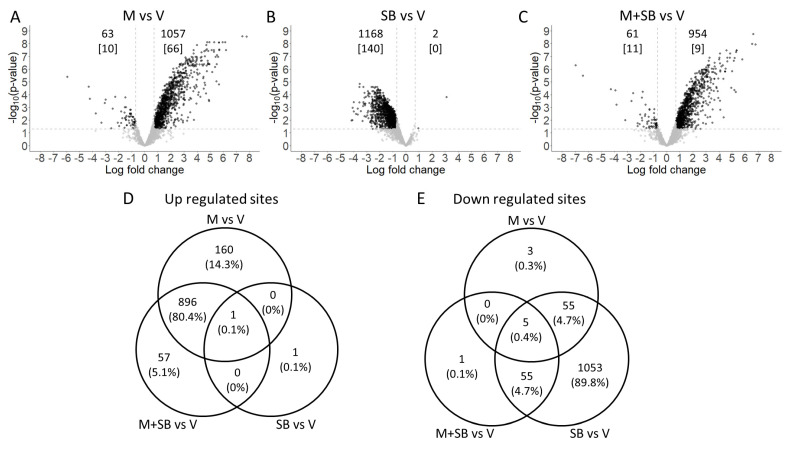
Phosphoproteomic analysis in HepaRG cells after treatment with MCLR and/or SB203580 (SB). HepaRG cells were treated with 50 μM SB or vehicle (V) for 1 h and then with 1 μM MCLR (M) or V for 24 h (*n* = 3). Volcano plots show increased and decreased mono-phosphosites (top number). The number of increased and decreased di-phosphorylated phosphosites are shown in brackets (data points not included in volcano plots). Horizontal dotted lines indicate 1.3 log_10_ *p*-value cut off and vertical dotted lines indicate ± 0.7 log2 fold change cutoff values used (**A**–**C**). Venn diagrams indicate number of shared phosphosites (percent in parentheses) between the different treatment groups (**D**,**E**).

**Figure 3 ijms-24-11168-f003:**
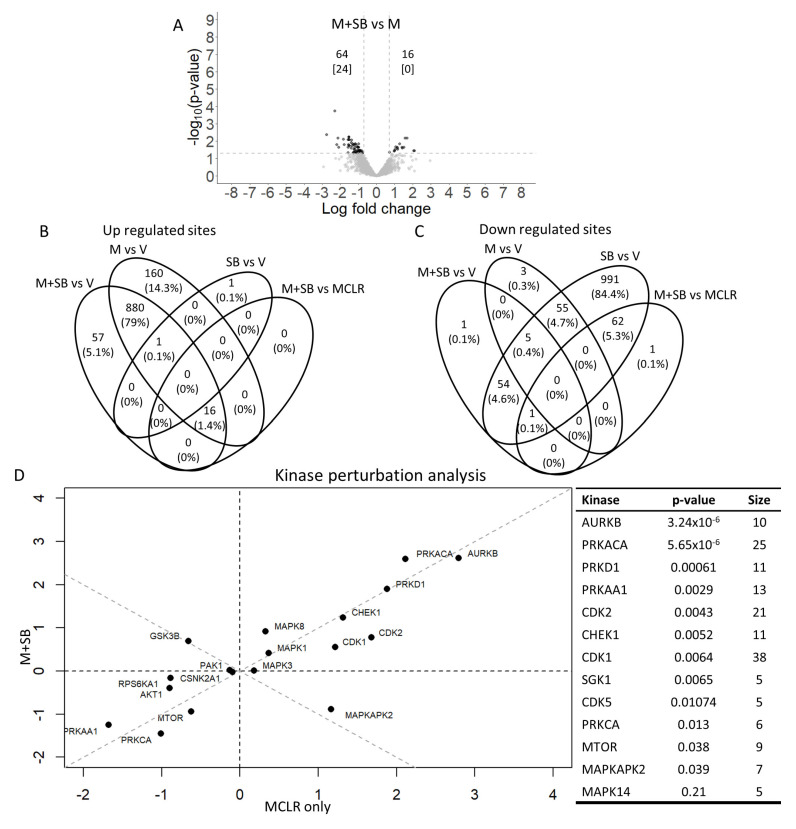
Phosphosite and kinase analysis of SB203580 (SB) rescue of MCLR toxicity. HepaRG cells were treated with 50 μM SB or vehicle (V) for 1 h and with 1 μM MCLR (M) or V for 24 h (*n* = 3). Volcano plot shows increased and decreased mono-phosphosites (top number). The number of increased and decreased di-phosphorylated phosphosites are shown in brackets (data points not included in volcano plot). Horizontal dotted lines indicate 1.3 log_10_ *p*-value cut off and vertical dotted lines indicate ± 0.7 log2 fold change cutoff values used (**A**). Venn diagrams indicate number of shared phosphosites (percent in parentheses) between the different treatment groups (**B**,**C**). Kinase perturbation analysis (**D**).

**Figure 4 ijms-24-11168-f004:**
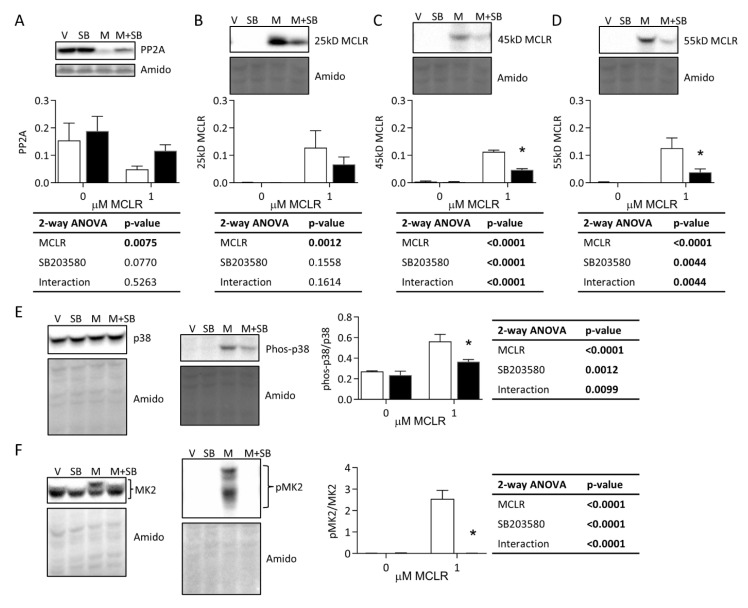
Western blot analysis of PP2A, MCLR, p38, and MK2. HepaRG cells were treated with 50 μM SB (black bars) or vehicle (V) (white bars) for 1 h and with or without 1 μM MCLR (M) for 24 h. Representative blots for PP2A (**A**), MCLR (**B**–**D**), total and phospho-p38 (**E**), and total and phospho-MK2 (**F**). Amido black total protein stain and 2-way ANOVA tables are shown for each panel. Error bars represent mean ± standard deviation of *n* = 3. * *p* < 0.05 represent Sidak’s multiple comparison post-test comparing vehicle treatment to SB treatment within each MCLR treatment group (0 and 1 µM).

**Figure 5 ijms-24-11168-f005:**
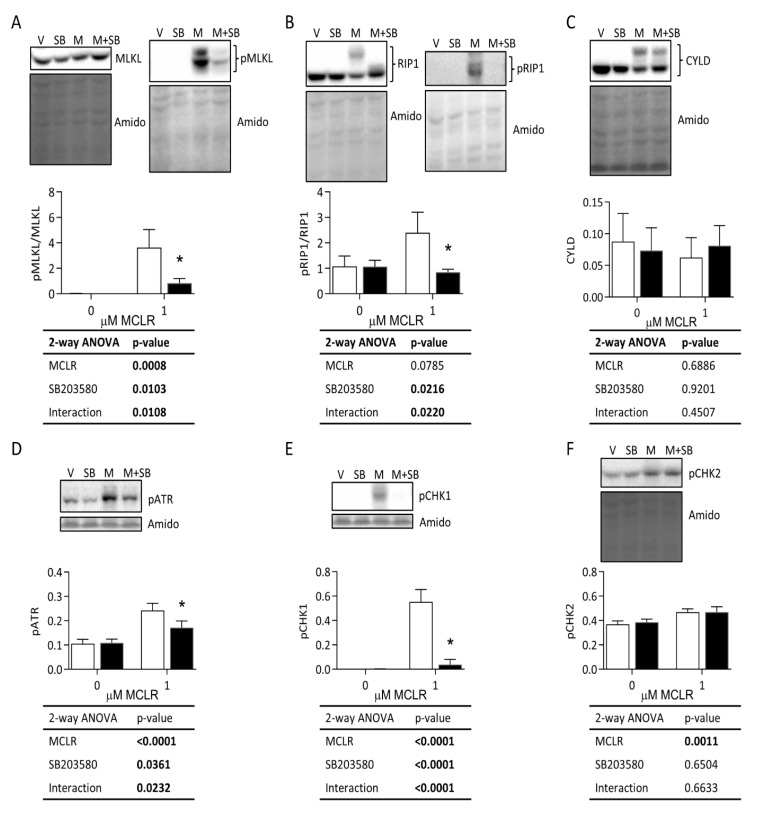
Western blot analysis of necroptosis and DNA damage markers. HepaRG cells were treated with 50 μM SB (black bars) or vehicle (V) (white bars) for 1 h and with or without 1 μM MCLR (M) for 24 h. Representative blots for total and phospho-MLKL (**A**), total and phospho-RIP1 (**B**), CYLD (**C**), phospho-ATR (**D**), phospho-CHK1 (**E**), and phospho-CHK2 (**F**). Amido black total protein stain and 2-way ANOVA tables are shown for each panel. Error bars represent mean ± standard deviation of *n* = 3. * *p* < 0.05 represent Sidak’s multiple comparison post-test comparing vehicle treatment to SB treatment within each MCLR treatment group (0 and 1 µM).

**Figure 6 ijms-24-11168-f006:**
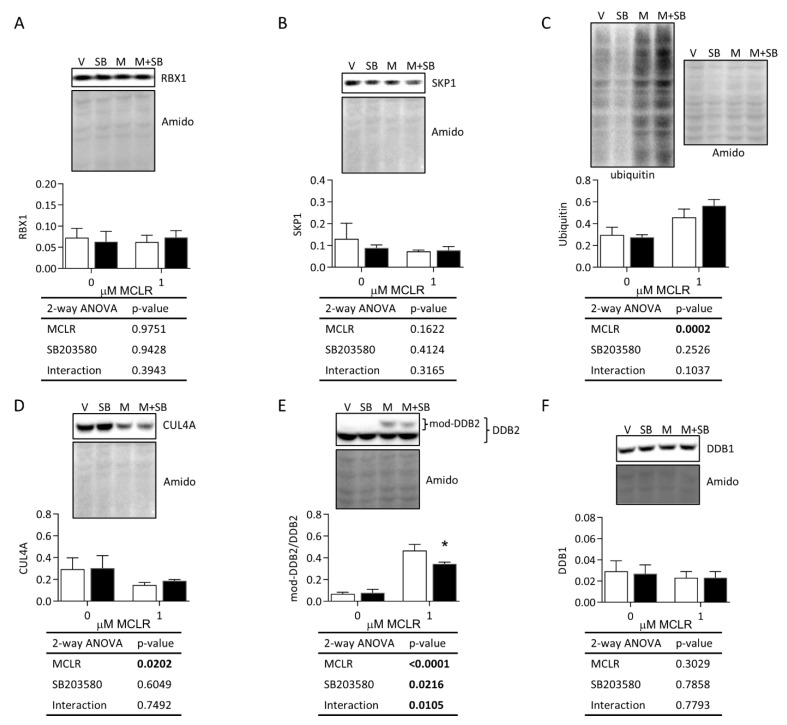
Western blot analysis of ubiquitination and DNA damage markers. HepaRG cells were treated with 50 μM SB (black bars) or vehicle (V) (white bars) for 1 h and with or without 1 μM MCLR (M) for 24 h. Representative blots for RBX1 (**A**), SKP1 (**B**), ubiquitin (**C**), CUL4A (**D**), DDB2 (**E**), and DDB1 (**F**). Amido black total protein stain and 2-way ANOVA tables are shown for each panel. Error bars represent mean ± standard deviation of *n* = 3. * *p* < 0.05 represent Sidak’s multiple comparison post-test comparing vehicle treatment to SB treatment within each MCLR treatment group (0 and 1 µM).

**Table 1 ijms-24-11168-t001:** Reactome pathways dysregulated by MCLR or SB203580 treatment.

MCLR Reactome	*p*-Value	Number of Substrates	SB203580 Reactome	*p*-Value	Number of Substrates
Activation of anterior HOX genes in hindbrain development during early embryogenesis	0.020	9	Activation of anterior HOX genes in hindbrain development during early embryogenesis	0.029	9
Activation of HOX genes during differentiation	0.020	9	Activation of HOX genes during differentiation	0.029	9
Apoptosis	0.038	34	Activation of the AP-1 family of transcription factors	0.043	5
Apoptotic cleavage of cellular proteins	0.035	16	Asparagine N-linked glycosylation	0.026	35
Apoptotic execution phase	0.011	18	Aspirin ADME	0.043	4
Carnitine metabolism	0.045	5	Chromosome maintenance	0.045	11
Cell junction organization	0.027	19	COPI-dependent Golgi-to-ER retrograde traffic	0.024	16
Condensation of prophase chromosomes	0.019	4	Diseases of glycosylation	0.032	5
COPII-mediated vesicle transport	0.049	12	DNA damage bypass	0.018	8
Cytoprotection by HMOX1	0.049	12	ER-to-Golgi anterograde transport	0.047	28
Depolymerization of the nuclear lamina	0.015	9	Gap junction trafficking	0.037	4
Disassembly of the destruction complex and recruitment of AXIN to the membrane	0.005	4	Gap junction trafficking and regulation	0.012	5
ERK/MAPK targets	0.019	6	Gluconeogenesis	0.000	7
Estrogen-dependent gene expression	0.048	18	Glycogen synthesis	0.012	4
Gene and protein expression by JAK-STAT signaling after interleukin-12 stimulation	0.043	8	Hemostasis	0.007	73
Glycogen breakdown (glycogenolysis)	0.031	5	Kinesins	0.039	9
Glycogen metabolism	0.045	7	L1CAM interactions	0.017	20
Initiation of nuclear envelope (NE) reformation	0.023	8	MAPK targets/nuclear events mediated by MAP kinases	0.009	8
Inositol phosphate metabolism	0.008	4	Membrane trafficking	0.026	123
Interleukin-12 family signaling	0.016	12	Metabolism of carbohydrates	0.022	38
Interleukin-12 signaling	0.043	8	Mitochondrial biogenesis	0.048	16
IRE1alpha activates chaperones	0.028	10	NOTCH3 activation and transmission of signal to the nucleus	0.046	5
Meiosis	0.035	8	Nuclear Events (kinase and transcription factor activation)	0.019	13
Meiotic synapsis	0.035	8	Peptide hormone metabolism	0.047	4
Metabolism of carbohydrates	0.010	38	Platelet activation, signaling, and aggregation	0.032	34
Muscle contraction	0.002	20	Platelet degranulation	0.002	17
Platelet degranulation	0.038	17	Post-translational protein phosphorylation	0.044	13
Platelet homeostasis	0.040	10	Regulation of beta-cell development	0.021	4
PPARA activates gene expression	0.009	18	Regulation of insulin-like growth factor (IGF) transport and uptake by insulin-like growth factor binding proteins (IGFBPs)	0.044	13
Recycling pathway of L1	0.007	9	Regulation of TP53 degradation	0.013	8
Regulation of glycolysis by fructose 2,6-bisphosphate metabolism	0.026	4	Regulation of TP53 expression and degradation	0.013	8
Regulation of lipid metabolism by PPARalpha	0.005	19	Response to elevated platelet cytosolic Ca^2+^	0.003	18
Regulation of TP53 activity through acetylation	0.021	7	RHOU GTPase cycle	0.045	17
Reproduction	0.035	8	Signaling by ERBB4	0.036	8
Response to elevated platelet cytosolic Ca^2+^	0.026	18	Signaling by NTRK1 (TRKA)	0.029	25
RHO GTPases activate IQGAPs	0.033	7	Signaling by NTRKs	0.016	27
RHOD GTPase cycle	0.004	19	Termination of translesion DNA synthesis	0.027	4
RHOF GTPase cycle	0.006	16	Toll-like receptor 4 (TLR4) cascade	0.043	16
RORA activates gene expression	0.046	5	TP53 regulates transcription of several additional cell death genes whose specific roles in p53-dependent apoptosis remain uncertain	0.036	4
SARS-CoV-2 activates/modulates innate and adaptive immune responses	0.034	22	Transport to the Golgi and subsequent modification	0.047	28
Signaling by NOTCH1	0.046	11	Vesicle-mediated transport	0.025	127
Smooth muscle contraction	0.015	11			
Sphingolipid metabolism	0.030	8			
Synthesis of bile acids and bile salts via 7alpha-hydroxycholesterol	0.029	4			
Transcriptional regulation of white adipocyte differentiation	0.032	13			
Transport of small molecules	0.025	63			
Unfolded protein response (UPR)	0.019	14			
XBP1(S) activates chaperone genes	0.028	10			

**Table 2 ijms-24-11168-t002:** Phosphosites dysregulated by MCLR with SB203580 versus MCLR alone.

Upregulated				Downregulated															
Gene	Site	LogFC	A.*p*.Val	Gene	Site	LogFC	A.P.Val	Gene	Site	LogFC	A.P.Val	Gene	Site	LogFC	A.P.Val	Gene	Site	LogFC	A.P.Val
AAK1	456	1.50	0.023	ANKS6	20	−1.00	0.033	EPB41L1	510	−1.24	0.044	LOC113230	233	−1.53	0.006	RBM15	612	−1.19	0.014
AAK1	249	1.14	0.023	BAG6	994	−2.77	0.004	EPB41L1	783	−1.07	0.045	MAP4K4	981	−0.97	0.046	RBM15	656	−1.00	0.014
BET1L	9	0.96	0.038	BAG6	1003	−1.53	0.006	F11R	264	−1.28	0.016	MTSS1L	455	−1.56	0.016	RPUSD2	35	−1.26	0.016
CDC42BPA	735	1.01	0.031	BSG	166	−1.57	0.008	FAM21C	245	−1.41	0.025	NA	308	−0.79	0.049	SCRIB	1486	−1.29	0.046
CDK11B	234	0.71	0.044	CLN6	7	−0.96	0.049	GPALPP1	103	−1.25	0.029	NKTR	79	−1.53	0.017	SCRIB	1475	−1.05	0.045
CLASRP	285	1.41	0.025	CUL4A	10	−1.32	0.014	GPRC5C	29	−0.98	0.036	NOLC1	707	−0.92	0.033	SCRIB	1378	−1.58	0.046
CTDP1	672	1.14	0.025	DDB2	26	−1.39	0.009	HDLBP	31	−2.31	0.0002	PARD3B	924	−1.52	0.019	SLC4A1AP	258	−1.09	0.019
DDX52	22	1.69	0.007	DDB2	24	−1.41	0.013	HNRNPM	481	−1.36	0.019	PHRF1	1202	−1.23	0.046	SMARCA2	1377	−0.77	0.047
ELMSAN1	461	2.09	0.035	DDX46	804	−0.95	0.036	HNRNPM	575	−1.57	0.023	PHRF1	936	−1.10	0.047	SPAG9	203	−1.05	0.023
EPB41L3	448	1.16	0.028	DLG1	586	−0.98	0.035	HNRNPM	452	−2.09	0.023	PHRF1	867	−1.10	0.046	SPTAN1	1197	−0.86	0.045
GPRC5C	68	0.98	0.035	EEF1D	162	−1.06	0.023	HNRNPM	513	−1.00	0.023	PHRF1	991	−1.27	0.049	SYNE1	7900	−1.21	0.037
HNRNPM	528	1.04	0.022	EFR3A	721	−1.06	0.049	HNRNPM	432	−1.00	0.023	PIK3C2A	338	−0.87	0.038	USP36	742	−2.13	0.007
SLC26A6	644	2.04	0.036	EIF4G3	494	−0.92	0.033	HNRNPM	86	−1.20	0.023	PIK3C2A	259	−0.91	0.038	VGLL4	59	−1.53	0.006
SRCIN1	1021	1.39	0.023	EIF5B	113	−1.83	0.008	HNRNPUL1	94	−1.20	0.029	PKP4	314	−1.29	0.046	VPS11	803	−1.79	0.016
ZMYND8	695	1.27	0.014	EIF5B	164	−1.53	0.008	HSPH1	73	−1.07	0.033	PPAN	306	−0.82	0.035	ZCCHC17	52	−2.20	0.016
ZNF446	3	1.57	0.007	EIF5B	214	−1.52	0.008	IQSEC1	166	−1.17	0.014	RAB11FIP5	307	−1.13	0.047	ZSCAN29	153	−1.13	0.037

Gene = Gene symbol, Site = phosphorylation site in protein sequence, LogFC = log fold change, A.P.Val = adjusted *p*-value, *n* = 3.

## Data Availability

The data presented in this study are openly available in FigShare at https://doi.org/10.6084/m9.figshare.23620539.v1 (accessed on 16 December 2022).

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
