# Peer review of "Involvement of the p38/MK2 Pathway in MCLR Hepatotoxicity Revealed through MAPK Pharmacological Inhibition and Phosphoproteomics in HepaRG Cells"

_ijms, 2023, doi:10.3390/ijms241311168_

Round 1

Reviewer 1 Report

Journal: IJMS (ISSN 1422-0067)

Manuscript ID: ijms-2396660

Type: Article

Title: Involvement of the p38/MK2 pathway in MCLR hepatotoxicity revealed through MAPK pharmacological inhibition and phosphoproteomics in HepaRG cells

Section: Molecular Toxicology

Special Issue: Molecular Mechanisms of Hepatotoxicity 2.0

This manuscript used specific pharmacological MAPK inhibitors were in HepaRG cells, a versatile human liver cell line that is functionally similar to primary hepatocytes, to examine the MAPK pathways associated with MCLR toxicity. In addition, phosphoproteomic analysis was performed after MCLR exposure in the presence and absence of a MAPK inhibitor to elucidate specific pathways important for MCLR-mediated hepatocyte death. However, the presentation is not so good. Also, deep analyses of phosphoproteomics are lacking. I have the following comments and suggestions for the authors to improve the quality of manuscript.

1. Lines 37-39

“MCLR exposures occur through drinking water, recreation in water, and/or consumption of contaminated fish and vegetables.1–3”

Please read and insert the following reference.

Challenges of using blooms of Microcystis spp. in animal feeds: A comprehensive review of nutritional, toxicological and microbial health evaluation. https://doi.org/10.1016/j.scitotenv.2020.142319

2. Lines 61-63

“For example, MCLR decreased PP2A activity and increased phosphorylation of AKT, ERK1/2, JNK, and p38 in mouse livers as early as 2 h post-exposure.”

Please insert the reference(s).

3. Lines 230-233

“MCLR decreased cell viability in a dose and time dependent manner (Figure 1A). The lowest toxic dose at 24 h was 1 µM, which decreased to 0.01 µM at 48 h and 72 h. MCLR doses at or above 30 µM decreased viability by greater than 50%.”

Please calculate the medium effective concentrations (EC50) at 24, 48 and 72 h, respectively.

4. Insufficient information is given in the material and methods section. More information about experimental design and treatment should be explained.

For example, the methods of treatment of N-acetyl-cystine (NAC), SP600125 (SP) MK2206 (MK) or SB203580 (SB) were not introduced.

5. Figure 1B-G

What is the time of treatment of N-acetyl-cystine (NAC), SP600125 (SP) MK2206 (MK) or SB203580 (SB)? More details are needed in the revised manuscript.

6. Figure 2

What are the vales in the “[]”, for example 10 and 66 in Fig. 2A. Please insert some text for descriptions in the revised manuscript.

7. Lines 264-266

“Among the down-regulated mono-phosphosites across all treatments, 94.7% were shared between the two SB treatments (i.e., SB alone and MCLR with SB) (Figure 2E)”

The data in Figure 2E showed 4.7% were shared between the two SB treatments, not 94.7%.

8. Similar to Table 1, please list the 1120 phosphosites dysregulated by MCLR, 1170 phosphosites dysregulated by SB203580 (SB) or 1015 phosphosites dysregulated by combination of MCLR/SB in the supplementary file. Then readers can better understand your article.

9. Section “3.3.1. Protein phosphatase expression, MCLR protein binding, and p38/MK2 phosphorylation”

“MCLR decreased PP2A protein expression and increased PP2A bound to MCLR, but SB did not rescue either MCLR effect (Figure 4A-B).”

How did you determine the binding of PP2A to MCLR? Please insert some explanations in the revised manuscript.

10. Figure 4B-D

What are MCLR-bound proteins at 25 kDa, 45 kDa and 55 kDa? More details are needed in the revised manuscript.

11. Figure 4A-F, Figure 5A-F, Figure 6A-F

Please use different letters to show significant differences among the four treatment groups. For example, "a" and "b" or "bc" or "c" have significant differences, but "b" and "ab" or "bc" have no significant differences. This expression method is better than by marker of “*”.Then the results of statistical analyses between control and MC-LR group can be shown readers can better understand your article.

12. Discussion

“Oxidative stress and DNA damage are interrelated mechanisms that are often implicated in MCLR hepatotoxicity, both of which can also activate the p38 pathway.15–18”

References 15-18 are not about MCs. Please insert some papers about MCs. For example,

Mechanisms of Microcystin-induced Cytotoxicity and Apoptosis. https://dx.doi.org/10.2174/1389557516666160219130407

Molecular mechanisms of microcystin toxicity in animal cells. https://doi.org/10.3390/ijms11010268

13. Graphical Abstract

Please insert this figure as Figure 7, the last figure for summarizing data of this manuscript.

14. This study performed phosphoproteomics. But results of phosphoproteomics were not fully described, explained and discussed. Many phosphosites of proteins were dysregulated by MCLR, SB203580 (SB) or combination of MCLR/SB, but only limited proteins and phosphosites were described in the text. In fact, the proteomics including phosphoproteomics can provide a global understanding of cellular regulation, including signaling pathways. Please search papers about proteomics and add some bioinformatic analyses.

ok

Reviewer 2 Report

In this paper, the authors have discussed the involvement of p38/MK2, DNA damage, and necroptosis pathways via MCLR-induced toxicity in HepaRG cells. The following changes to improve the manuscript.
1. Please include a heading/title for graphs in Fig 1 for clarity;
2. Fig 1 - E,F,G: Why does the percent viability of the Vehicle (dotted line) cells seems to be decreasing significantly compared to those treated with MCLR and SB? However, this is not the case in Fig 1 B,C,D graphs. Please explain;
3. The authors say, 'Among the down-regulated mono-phosphosites across all treatments, 94.7% were shared between the two SB treatments', but it does not correlate in the graphs 2B and 2C. Please tabulate all of the up vs down-regulated genes/mono-phosphosites separately for graphs in Fig 2A, B and C for clarity;
4. Spell check: Line 260 - Venn diagram instead of Ven.

Minor spell check 

Reviewer 3 Report

The manuscript by Lynch and colleagues goes into the deep of the mechanism underlying the toxicity of MCLR in liver-derived cells of human origin. The investigation relies on the analysis of the phosphoproteome of HepaRG cells treated with MCLR either in the absence of presence of the specific p38 inhibitor SB203580. The results reveal that SB203580 was capable of preventing MCLR toxicity and of reducing the phosphorylation of a number of proteins controlling the execution of the necroptotic program of cell death. By contrast, NAC, a hydrophilic ROS scavenger and GSH-inducing agent, was ineffective in protecting the cells against MCLR toxicity. The Authors conclude that MCLR likely triggers necroptosis in HepaRG cells, and that oxidative stress is not partaking in the death mechanism.

The research is interesting, the cellular model adequate; the methods are of high quality and well executed. The results are basically well and clearly presented and support most of the conclusions of the Authors. In spite of the overall appreciation of the research, the few aspects listed below deserve attention to further strengthen the results.

Major criticisms

1. Occurrence of necroptosis in MCLR toxicity in HepaRG cells. I agree with the Authors’ view that MCLR does not primarily modulate the amount of proteins involved in necroptosis. However, in the present research, occurrence of necroptosis is only speculative and not verified, as the Authors correctly state in the Discussion (‘…that necroptosis may be an important aspect of MCLR-mediated hepatotoxicity.’). Occurrence of this type of cell death must be confirmed in this experimental model and settings using classical necroptosis inhibitors (different from p38 inhibitors).

2. The role of oxidative stress in MCLR toxicity is a critical point. The lack of effect of NAC in preventing MCLR-induced toxicity in HepaRG cells suggests that oxidative stress is not involved. However, it is well known that not all oxidant species are effectively buffered and neutralized by hydrophilic ROS scavengers, as NAC is. Also, alternative possibilities and opposite demonstrations exit in literature, as correctly reported by the Authors. Eventually, it has to be kept in consideration that most frequently, yet not exclusively, DNA damage consequent to toxins (as it is postulated to occur in this research) or to drugs relies on an oxidative stress- based mechanism. Thus, to definitely prove the Authors’ hypothesis, at least one lipophilic antioxidant should be used, coupled to a measurement of ROS generation under the standard treatment conditions with MCLR.

Minor

Lines 106-107. Seeding of cells was done at 9,000/well in a 96-well plate or 200,000/well in 12-well plate, which makes a density of about 27000 and 50,000 cells/cm2, respectively. Is there any special reason for not keeping constant the cell density (and the ratio of MCLR molecules/cells) over different cell culture supports?

Round 2

Reviewer 1 Report

Journal: IJMS (ISSN 1422-0067)

Manuscript ID: ijms-2396660-peer-review-v2

Type: Article

Title: Involvement of the p38/MK2 pathway in MCLR hepatotoxicity revealed through MAPK pharmacological inhibition and phosphoproteomics in HepaRG cells

Section: Molecular Toxicology

Special Issue: Molecular Mechanisms of Hepatotoxicity 2.0

This manuscript used specific pharmacological MAPK inhibitors were in HepaRG cells, a versatile human liver cell line that is functionally similar to primary hepatocytes, to examine the MAPK pathways associated with MCLR toxicity. In addition, phosphoproteomic analysis was performed after MCLR exposure in the presence and absence of a MAPK inhibitor to elucidate specific pathways important for MCLR-mediated hepatocyte death.

The manuscript improved during the revision. However, there are still some issues need to be addressed. I have the following comments and suggestions for the authors to improve the quality of manuscript.

1. Similar to Supplementary Table 1, please list the increased and decreased di-phosphorylated phosphosites dysregulated by MCLR, SB203580 (SB) or combination of MCLR/SB in the supplementary file. Then readers can better understand your article.

2. Are the present analyses only about mono-phosphosites? Please state this clearly, expecially in the figure legends.

3. Please add data of analyses of di-phosphorylated phosphosites.

4. Figure 4A-F, Figure 5A-F, Figure 6A-F

Please use different letters to show significant differences among the four treatment groups. For example, "a" and "b" or "bc" or "c" have significant differences, but "b" and "ab" or "bc" have no significant differences. This expression method is better than by marker of “*”. Then the results of statistical analyses between control and MC-LR group can be shown, and readers can better understand your article.

I know the advantages of two-way ANOVA. So keep the results of two-way ANOVA in the table. Please insert data of one-way ANOVA, by different letters in the figures.

5. Figure 4C-D

Lines 265-269

“MCLR decreased PP2A protein expression and increased protein bound MCLR, but SB did not significantly rescue either MCLR effect (Figure 4A-B). In contrast, two higher molecular weight MCLR-bound proteins at 45 kDa and 55 kDa were partially rescued by SB (Figure 4C-D). The MCLR-bound proteins, represented by these different molecule weights, were not determined here”

“The MCLR-bound proteins, represented by these different molecule weights, were not determined here”, what do you mean? It is unclear.

6. Figure 4C-D

What are MCLR-bound proteins at 25 kDa, 45 kDa and 55 kDa? More details are needed in the revised manuscript.

7. Lines 384-385

“Although many publications cite oxidative stress as an important mechanism in MCLR toxicity”

Lines 397-398

“although oxidative stress may be involved in MCLR-mediated cell death”

Please read and cite the following paper.

Mechanisms of Microcystin-induced Cytotoxicity and Apoptosis.

https://dx.doi.org/10.2174/1389557516666160219130407

https://www.researchgate.net/publication/295246594_Mechanisms_of_Microcystin-induced_Cytotoxicity_and_Apoptosis

You can get the paper by ResearchGate.

8. Graphical Abstract

The journal Toxins does not publish Graphical Abstract in the pdf version. Please insert this figure as Figure 7, the last figure for summarizing data of this manuscript. Then readers can better understand your article.

ok

Author Response

All changes to the manuscript text are highlighted in yellow in the revised manuscript document.

Point 1: Similar to Supplementary Table 1, please list the increased and decreased di-phosphorylated phosphosites dysregulated by MCLR, SB203580 (SB) or combination of MCLR/SB in the supplementary file. Then readers can better understand your article.

Response 1: Thanks for suggestion. The di-phosphosites have now been included as Supplemental Table 2.

Point 2: Are the present analyses only about mono-phosphosites? Please state this clearly, expecially in the figure legends.

Response 2: The distinction between analyses related to mono-phosphosites versus di-phosphosites are clearly differentiated in manuscript. We have highlighted these places for the reviewer.  

Point 3: Please add data of analyses of di-phosphorylated phosphosites.

Response 3: We have now included the same data analyses for di-phosphosites that were completed for the mono-phosphosites. The overlapping di-phosphosites are discussed in the Results section 3.1 and no Venn diagrams are needed because the data are simple enough to be covered in the text. No Reactome pathways were identified in this small di-phosphosite dataset.

Point 4: Figure 4A-F, Figure 5A-F, Figure 6A-F

Please use different letters to show significant differences among the four treatment groups. For example, "a" and "b" or "bc" or "c" have significant differences, but "b" and "ab" or "bc" have no significant differences. This expression method is better than by marker of “*”. Then the results of statistical analyses between control and MC-LR group can be shown, and readers can better understand your article.

I know the advantages of two-way ANOVA. So keep the results of two-way ANOVA in the table. Please insert data of one-way ANOVA, by different letters in the figures.

Response 4: We respectfully disagree with the reviewer’s suggestion. It is not appropriate to perform both two-way ANOVA and one-way ANOVA on the same dataset, especially because two-way ANOVA should be used when there are two factors in the study design (i.e., MCLR treatment and SB treatment). The “statistical analysis between control and MC-LR group” requested by the reviewer is captured in the two-way ANOVA p-value for MCLR.

Point 5: Figure 4C-D

Lines 265-269

“MCLR decreased PP2A protein expression and increased protein bound MCLR, but SB did not significantly rescue either MCLR effect (Figure 4A-B). In contrast, two higher molecular weight MCLR-bound proteins at 45 kDa and 55 kDa were partially rescued by SB (Figure 4C-D). The MCLR-bound proteins, represented by these different molecule weights, were not determined here”

“The MCLR-bound proteins, represented by these different molecule weights, were not determined here”, what do you mean? It is unclear.

Response 5: Thank you for asking for clarification. This analysis is a western blot using an MCLR antibody. Therefore, the bands that were identified at 25 kDa, 45 kDa, and 55 kDa represent specific proteins that have MCLR covalently bound to them. The revised manuscript states that we did not determine what proteins are bound to MCLR. It is beyond the scope of this manuscript to identify what the MCLR bands represent. The results have been carefully written to clearly state that we have not identified what these bands represent.

Point 6: Figure 4C-D

What are MCLR-bound proteins at 25 kDa, 45 kDa and 55 kDa? More details are needed in the revised manuscript.

Response 6: This point made by the reviewer addresses the same point raised in Point 5. We do not know what the 25 kDa, 45 kDa, and 55 kDa bands detected by the MCLR antibody represent. This manuscript is focused on phosphoproteomics and molecular pathways associated with MCLR hepatotoxicity. Previous publications have already identified several proteins to which MCLR covalently binds (i.e., PP1, PP2A, and ATP synthase). It is outside the scope of this manuscript to characterize the proteins bound by MCLR.

Point 7: Lines 384-385

“Although many publications cite oxidative stress as an important mechanism in MCLR toxicity”

Lines 397-398

“although oxidative stress may be involved in MCLR-mediated cell death”

Please read and cite the following paper. Mechanisms of Microcystin-induced Cytotoxicity and Apoptosis. https://dx.doi.org/10.2174/1389557516666160219130407

https://www.researchgate.net/publication/295246594_Mechanisms_of_Microcystin-induced_Cytotoxicity_and_Apoptosis

You can get the paper by ResearchGate.

Response 7: The reference has been added.

Point 8: Graphical Abstract

The journal Toxins does not publish Graphical Abstract in the pdf version. Please insert this figure as Figure 7, the last figure for summarizing data of this manuscript. Then readers can better understand your article.

Response 8: The journal is International Journal of Molecular Sciences, not Toxins. When we inquired about the reviewer’s request, the Assistant Editor stated this, “According to the journal's rule, the graphical abstract cannot be the same as the in-text figure.” Thus, we cannot include the graphical abstract as a figure because the journal prohibits it.

Reviewer 3 Report

In my opinion, the manuscript by Lynch and colleagues has not improved from the experimental point of view. The Authors have further detailed in the discussion the controversial points without providing any additional demonstration of the occurrence of necroptosis and of oxidative stress. Since the Authors agree, and state in the Discussion, that these aspects would deserve consideration, it is not clear why they did not perform the simple and quick experiments that would have definitely solved these problems. In addition, in their reply the Authors generically state that they ‘are unable to perform another set of experiments…’, but do not provide any valid explanation for why they cannot perform these very simple and quick experiments.

Author Response

Reviewer comment: In my opinion, the manuscript by Lynch and colleagues has not improved from the experimental point of view. The Authors have further detailed in the discussion the controversial points without providing any additional demonstration of the occurrence of necroptosis and of oxidative stress. Since the Authors agree, and state in the Discussion, that these aspects would deserve consideration, it is not clear why they did not perform the simple and quick experiments that would have definitely solved these problems. In addition, in their reply the Authors generically state that they ‘are unable to perform another set of experiments…’, but do not provide any valid explanation for why they cannot perform these very simple and quick experiments.

Response: We sincerely appreciate the reviewer for their desire for mechanistic information and clarity in discussing these points during peer review. We share these feelings and appreciate the opportunity to clarify. We apologize for not clearly stating the reasons additional experiments cannot be performed for this manuscript. The journal only provided 10 calendar days for the revisions. It would be difficult to complete even simple and quick experiments in that timeframe. In addition, working with HepaRG cells is neither simple nor quick. It would require > 2 month to get the cells ready for treatment due to the need for expansion and differentiation, and > 1 month would be needed to perform multiple optimization experiments for each additional treatment compound added into the experimental design. This timeline also assumes the experiments work the first time. Because we currently do not have the personnel time to complete the requested experiments, we have taken exceptional care not to overstate our results in the revised manuscript. We are open to further suggestions from the reviewer for how to provide appropriate interpretations of our results without additional experiments.

We hope that the reviewer will consider the novelty of this manuscript in its current form. For example, this is the first human liver phosphoproteomic analysis after MCLR exposure. Considering the fact that MCLR directly and indirectly affects phosphoproteins by inhibiting protein phosphatases and activating MAPKs, respectively, this work will be valuable to the cyanotoxin research community. We will also add important knowledge regarding which MAPK may be important for human hepatocyte death, and we provide the first phosphoproteomic analysis for how a p38 inhibitor modifies the MCLR-elicited phosphoproteome. We demonstrate that phospho-MLKL increased with MCLR treatment and was reversed by p38 inhibitor treatment. Importantly, phospho-MLKL levels is a recognized marker of necroptosis. Thus, although a necroptosis inhibitor would further confirm the results, the results as presented provide strong indication that necroptosis occurred in our experiments. In addition, NAC is the most common antioxidant used after MCLR toxicity, therefore by including it we have provided an important treatment group for comparison to other publications. The additional of lipophilic antioxidants is a clever suggestion that we will certainly consider with any future antioxidant studies we perform, but we believe it is not necessary to demonstrate that NAC did not rescue the MCLR toxicity. Again, we appreciate that the reviewer has provided meaningful suggestions, but hope the novelty of the work justifies publication in its current form.

Round 3

Reviewer 1 Report

Journal: IJMS (ISSN 1422-0067)

Manuscript ID: ijms-2396660-peer-review-v3

Type: Article

Title: Involvement of the p38/MK2 pathway in MCLR hepatotoxicity revealed through MAPK pharmacological inhibition and phosphoproteomics in HepaRG cells

Section: Molecular Toxicology

Special Issue: Molecular Mechanisms of Hepatotoxicity 2.0

This manuscript used specific pharmacological MAPK inhibitors were in HepaRG cells, a versatile human liver cell line that is functionally similar to primary hepatocytes, to examine the MAPK pathways associated with MCLR toxicity. In addition, phosphoproteomic analysis was performed after MCLR exposure in the presence and absence of a MAPK inhibitor to elucidate specific pathways important for MCLR-mediated hepatocyte death.

The manuscript improved during the revision. However, there are still some issues need to be addressed. I have the following comments and suggestions for the authors to improve the quality of manuscript.

1. Lines 81-82

“MCLR (cat. 10007188) was purchased from Cayman Chemicals (Ann Arbor, MI, USA).”

What was the purity of MC-LR? Did the Cayman Chemicals produce the MC-LR, or did they only sell MC-LR which was produced by other companies? Please present the information of company which produced the MC-LR and purity of MC-LR in the revised manuscript.

2. Undifferentiated HepaRG cells (passage 14) were purchased, and were cultured with Fully differential media. In some experiments, differentiated HepaRG cells were used. What are the differences between undifferentiated and differentiated HepaRG cells? Please insert some text to describe the differences. Which differentiated HepaRG cell types did you use? Also, please clearly write whether undifferentiated or differentiated HepaRG cells were exposed to MCLR in each sub-section of Materials and Methods, figure or table legends. This is very important information?

3. Fig. 2ACD

Number of up-regulated mono-phosphosites in the MCLR group was 1057 in Fig. 2A, but was 1056 in Fig. 2D. Number of up-regulated mono-phosphosites in the (MCLR and SB) group was 954 in Fig. 2C, but was 952 in Fig. 2D. Please carefully check your data.

4. Fig. 2ABE

Number of down-regulated mono-phosphosites in the MCLR group was 63 in Fig. 2A, but was 62 in Fig. 2E. Number of up-regulated mono-phosphosites in the SB group was 1168 in Fig. 2B, but was 1166 in Fig. 2E. Please carefully check your data.

5. Fig. 2AC, Fig. 3B

Number of up-regulated mono-phosphosites in the MCLR group was 1057 in Fig. 2A, but was 1056 in Fig. 3B. Number of up-regulated mono-phosphosites in the (MCLR and SB) group was 954 in Fig. 2C, but was 952 in Fig. 3B. Please carefully check your data.

6. Fig. 2AB, Fig. 3C

Number of down-regulated mono-phosphosites in the MCLR group was 63 in Fig. 2A, but was 62 in Fig. 3C. Number of up-regulated mono-phosphosites in the SB group was 1168 in Fig. 2B, but was 1166 in Fig. 3C. Please carefully check your data.

7. Lines 244-245

“All the dysregulated mono-phosphosites and di-phosphosites sites are listed in Supplemental Tables 1 and 2, respectively.”

There is only a table which shows di-phosphosites. Please insert the data of mono-phosphosites.

8. Supplemental Tables

Please also insert data of dysregulated mono-phosphosites and di-phosphosites sites for (MCLR and SB) group vs MCLR group.

9. Graphical Abstract

If the journal IJMS publish Graphical Abstract in the pdf version. Please insert this figure in the first page. I have never seen a journal which put the Graphical Abstract at the end of a paper.

10. References list

Please check the style of references for IJMS. The references are not in the right style.

ok

Author Response

Response to Reviewer 1 Comments- Round 3

All changes to the manuscript text are highlighted in yellow in the revised manuscript document.

Point 1: Lines 81-82

“MCLR (cat. 10007188) was purchased from Cayman Chemicals (Ann Arbor, MI, USA).”

What was the purity of MC-LR? Did the Cayman Chemicals produce the MC-LR, or did they only sell MC-LR which was produced by other companies? Please present the information of company which produced the MC-LR and purity of MC-LR in the revised manuscript..

Response 1: The MCLR purity is ≥95%. This information has now been included in the Methods section of the manuscript. Cayman Chemical Company is a reputable vendor, and they provide a Certificate of Analysis with every product. Providing the purity and the Cayman catalog number is sufficient information for a reader to be able to reproduce the experiments. At the reviewer’s request, we searched for the manufacturer’s product information, but Cayman does not specify where it was produced. Even if this product was manufactured by another company, it is unreasonable to request that original manufacturers be included in the manuscript because there are many vendors that sell products they do not manufacture. 

Point 2: Undifferentiated HepaRG cells (passage 14) were purchased, and were cultured with Fully differential media. In some experiments, differentiated HepaRG cells were used. What are the differences between undifferentiated and differentiated HepaRG cells? Please insert some text to describe the differences. Which differentiated HepaRG cell types did you use? Also, please clearly write whether undifferentiated or differentiated HepaRG cells were exposed to MCLR in each sub-section of Materials and Methods, figure or table legends. This is very important information?

Response 2: All experiments were completed using differentiated HepaRG. Section 2.2 in the methods has been modified to make this clearer. Differentiated HepaRG cells consist of two different cell phenotypes, biliary-like cells and hepatocytes. Both cell types are produced during differentiation and are not seeded separately. References have been added to the introduction that describes HepaRG characterization.

Point 3: Fig. 2ACD

Number of up-regulated mono-phosphosites in the MCLR group was 1057 in Fig. 2A, but was 1056 in Fig. 2D. Number of up-regulated mono-phosphosites in the (MCLR and SB) group was 954 in Fig. 2C, but was 952 in Fig. 2D. Please carefully check your data.

Response 3: Thank you for catching these small discrepancies. We have reviewed the data again and confirmed that the numbers in the Venn diagrams were off by a few numbers. We have changed the Venn diagrams to reflect the correct numbers that are shown in the volcano plots.

Point 4: Fig. 2ABE

Number of down-regulated mono-phosphosites in the MCLR group was 63 in Fig. 2A, but was 62 in Fig. 2E. Number of up-regulated mono-phosphosites in the SB group was 1168 in Fig. 2B, but was 1166 in Fig. 2E. Please carefully check your data.

Response 4: Same as response 3.

Point 5: Fig. 2AC, Fig. 3B

Number of up-regulated mono-phosphosites in the MCLR group was 1057 in Fig. 2A, but was 1056 in Fig. 3B. Number of up-regulated mono-phosphosites in the (MCLR and SB) group was 954 in Fig. 2C, but was 952 in Fig. 3B. Please carefully check your data.

Response 5: Same as response 3.

Point 6: Fig. 2AB, Fig. 3C

Number of down-regulated mono-phosphosites in the MCLR group was 63 in Fig. 2A, but was 62 in Fig. 3C. Number of up-regulated mono-phosphosites in the SB group was 1168 in Fig. 2B, but was 1166 in Fig. 3C. Please carefully check your data.

Response 6: Same as response 3.

Point 7: Lines 244-245

“All the dysregulated mono-phosphosites and di-phosphosites sites are listed in Supplemental Tables 1 and 2, respectively.”

There is only a table which shows di-phosphosites. Please insert the data of mono-phosphosites.

Response 7: Our apologies for this. It appears that the supplementary table 1 did not stay in the submission system during the revision process. To avoid this problem, we have combined the mono- and di-phosphosite data into one supplemental table.  

Point 8: Supplemental Tables

Please also insert data of dysregulated mono-phosphosites and di-phosphosites sites for (MCLR and SB) group vs MCLR group.

Response 8: The dysregulated mono-phosphosites for the MLCR and SB vs MCLR comparison are shown in Table 2. The dysregulated di-phosphosites for this comparison have been added to supplemental table 1.

Point 9: Graphical Abstract

If the journal IJMS publish Graphical Abstract in the pdf version. Please insert this figure in the first page. I have never seen a journal which put the Graphical Abstract at the end of a paper.

Response 9: We upload the graphical abstract completely separate from the manuscript file. We do not insert the graphical abstract into the manuscript file. It is inserted at that location in the manuscript file by the editor or submission system. We have no control over where it appears in the manuscript. The associate editor is aware that the graphical abstract does not go at the end of the manuscript, and it will not be published in that location. This appears to be a format used only during peer-review. 

Point 10: References list

Please check the style of references for IJMS. The references are not in the right style.

Response 10: The journal does not require a specific reference format. The Instructions for Authors states, “Your references may be in any style, provided that you use the consistent formatting throughout.”

Reviewer 3 Report

I have no furhter questions for the Authors.

Author Response

We thank reviewer 3 for their comments and appreciate their time for reviewing our manuscript.

Round 4

Reviewer 1 Report

Journal: IJMS (ISSN 1422-0067)

Manuscript ID: ijms-2396660-peer-review-v4

Type: Article

Title: Involvement of the p38/MK2 pathway in MCLR hepatotoxicity revealed through MAPK pharmacological inhibition and phosphoproteomics in HepaRG cells

Section: Molecular Toxicology

Special Issue: Molecular Mechanisms of Hepatotoxicity 2.0

The manuscript "Involvement of the p38/MK2 pathway in MCLR hepatotoxicity revealed through MAPK pharmacological inhibition and phosphoproteomics in HepaRG cells" evaluates the effects associated with MCLR cytotoxicity and a p38 inhibitors by phosphoproteomics. This study is innovative and makes an important contribution to the field. You have done a good piece of research which deserves publication. The manuscript improved a lot during the revisions. However, I have some specific comments and suggestions for the authors to consider:

1. An important issue is the process of extraction of proteins, which should be done with protease and phosphatase inhibitors for phosphoproteomic analysis, and if so, it needs to be clearly stated in "2.4 Immunoblot assays".

2. Line 150: Was it 40℃, or 4℃?

3. Why were the concentrations of NAC (0.5 mM or 5 mM), SP (25 μM or 50 μM), MK (0.1 μM or 1 μM) and SB (50 μM or 100 μM) different in section “3.1” (Fig. 1)? Please insert the reason why you chose different concentrations for NAC, SP, MK or SB in the viability assays in section “2.3. Viability assays”.

4. Please insert the tested MC-LR concentrations clearly in the legend of Fig 1A.

5. Lines 209-211

“SB completely rescued viability up to 10 µM MCLR at 24 and 48 h, however SB did not completely rescue viability at 72 h (Figure 1E-G).”

Fig. 1F shows that 50 µM SB did not completely rescue viability for 10 µM MCLR at 48 h. Please check it.

Author Response

Response to Reviewer 1 Comments- Round 4

All changes to the manuscript text are highlighted in yellow in the revised manuscript document.

Point 1: An important issue is the process of extraction of proteins, which should be done with protease and phosphatase inhibitors for phosphoproteomic analysis, and if so, it needs to be clearly stated in "2.4 Immunoblot assays"

Response 1: The protease and phosphatase inhibitor cocktail used in these experiments is listed in Section 2.1. We have highlighted it for the reviewer. As standard practice in molecular biology, it is included in the NP-40 lysis buffer.  

Point 2: Line 150: Was it 40℃, or 4℃?

Response 2: The typo has been corrected.

Point 3: Why were the concentrations of NAC (0.5 mM or 5 mM), SP (25 μM or 50 μM), MK (0.1 μM or 1 μM) and SB (50 μM or 100 μM) different in section “3.1” (Fig. 1)? Please insert the reason why you chose different concentrations for NAC, SP, MK or SB in the viability assays in section “2.3. Viability assays”

Response 3: The doses were selected to be pharmacologically relevant and to avoid cytotoxicity (see Section 2.3).     

Point 4: Please insert the tested MC-LR concentrations clearly in the legend of Fig 1A.

Response 4: The labels on the x-axis of Figure 1 A indicate MCLR concentrations. It would be redundant to include the concentrations in the figure legend.    

Point 5: Lines 209-211

“SB completely rescued viability up to 10 µM MCLR at 24 and 48 h, however SB did not completely rescue viability at 72 h (Figure 1E-G).”

Fig. 1F shows that 50 µM SB did not completely rescue viability for 10 µM MCLR at 48 h. Please check it.

Response 5: Both concentrations of SB significantly rescued HepaRG viability at 10 µM MCLR (see the asterisks in Figure 1F). We have added the qualifier that complete rescue occurs at 100 µM SB.